# Comparative Analysis and Determination of the Fatty Acid Composition of Kazakhstan's Commercial Vegetable Oils by GC-FID

Maxat Toishimanov [1,*] , Meruyet Nurgaliyeva [2,*], Assiya Serikbayeva [1], Zhulduz Suleimenova [1] , Karima Myrzabek [1], Aksholpan Shokan [3] and Nurgul Myrzabayeva [1]

1    Kazakh National Agrarian Research University, Almaty 050010, Kazakhstan
2    Kazakh Scientific Research Veterinary Institute, Almaty 050016, Kazakhstan
3    Al-Farabi Kazakh National University, Almaty 050040, Kazakhstan
*    Correspondence: maxat.toishimanov@gmail.com (M.T.); meruet79@gmail.com (M.N.)

**Abstract:** Here, we present the results of analyzing the fatty acid composition of the main edible vegetable oils from Kazakhstani oilseed producers (safflower, sunflower, maize (corn), linseed, cottonseed, soybean and rapeseed) in comparison with the known fatty acid (FA) composition of specific vegetable oils complying with the Codex Standard for Named Vegetable Oil (Codex Stan 210-1999). The fatty acid composition of 35 different vegetable oils was analyzed by gas chromatography with a Shimadzu GC-2010 Plus instrument with flame ionization detection using a high-polarity CP-Sil 2560, which allowed us to establish their authenticity for high accuracy and excellent separation. A comparative study of the fatty acid composition, groups and omega-6/omega-3 ratios in seven different vegetable oils was carried out. Subsequently, the data were processed by hierarchical clustering analysis, principal component analysis, artificial neural network and Pearson's correlation. Artificial neural network analysis demonstrated correct predictions. Principal component analysis showed the effects of oleic, linoleic and α-linolenic acids to vegetable oils classification.

**Keywords:** vegetable oil; authenticity; fatty acid; gas chromatography; validation; Kazakhstan

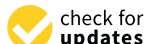



## 1. Introduction

Vegetable oils (VOs) are an important food product in the daily diet. As well as having a relatively low cost, they contain essential nutrients that provide many benefits, including reducing the risk of cardiovascular diseases, improving metabolism, and providing a person with energy and nutritional components [1–4].

Edible VOs are food products consisting primarily of the glycerides of fatty acids (FAs) and their accompanying substances, obtained exclusively from vegetable sources and containing at least 99% fat. The high biological value of VOs is due to the content of polyunsaturated FAs (linoleic acid C18:2 and linolenic acid C18:3), fat-soluble vitamins (A, D, E, K), phospholipids (lecithin) and carotenoids, including beta-carotene (provitamin A) [5–10].

In recent years, the relevance of a healthy lifestyle has increased. The World Health Organization recommends reducing certain types of fats and oils (butter, ghee and lard) in favor of healthier oils such as olive, soybean, rapeseed (canola), maize, safflower and sunflower oils [11].

Kazakhstan is located in the middle of the Eurasian continent. Because of its particular climatic and geographical situation, certain types of VOs grow in the country. Kazakhstan's fat and oil industry is actively increasing the production of oilseeds not only within the country, but also abroad, due to the diversity of the raw fat and oil materials cultivated for local conditions. The fat and oil market of the country can be divided into raw materials markets, primary processing products and final products [12].

Kazakhstan's fat and oil industry is focused on the use of domestic raw materials, primarily the production of sunflower oil. The climate and soil conditions of Kazakhstan allow producers to grow various oilseeds that can successfully compete in international markets. This applies to both the traditional crops of Kazakhstan, such as sunflower, cottonseed and linseed, and those that currently occupy small areas but are actively developing, such as soybeans, rapeseed, safflower and other crops [13]. In 2020, a total of 484.8 thousand tons of VOs were produced. The main VO was sunflower (320.1 thousand tons) and rapeseed (68.8 thousand tons) oils, which accounted for more than 80% of the total production. Nevertheless, soybean, safflower, linseed and maize oils production increased by 47%, 62%, 79% and 86% in five years, respectively [14] (Figure 1).

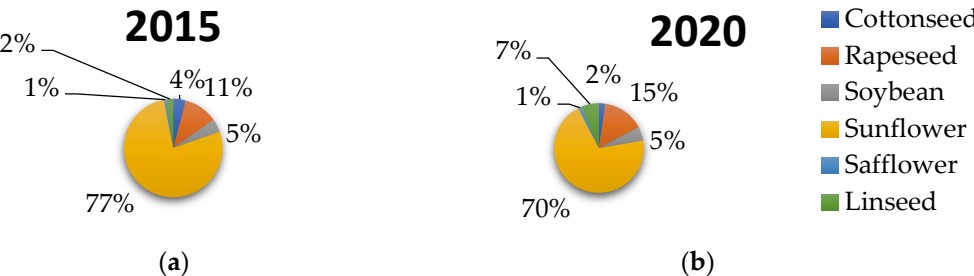

**Figure 1.** Share of VO types produced in Kazakhstan by year: (**a**) 2015 and (**b**) 2020. All data were obtained from the Bureau of National Statistics [14].

The domestic fat and oil market is diverse, and the volume of oil production is expanding, as a result of which, there is concern with regard to the authenticity of VOs. In addition, the market receives not only edible VOs, but also those technologically processed for food. The country's market is also filled with incoming imported fat and oil products. Given the rather complex chemical composition of VOs, there are problems with conducting a comprehensive examination of the authenticity of all types of VOs sold in the country's food markets [15].

A reliable way to determine the type of oil is to establish its chemical composition. The FA profile contains stable and informative characteristics by which the type of oil can be identified [16]. The analysis of FA profiles in food products requires the extraction of lipids from samples, followed by separation, identification and determination using various methods. Another important factor is the presence and ratio of the polyunsaturated FAs of the omega-6 and omega-3 groups, which perform vital functions and are part of the membrane structures of cells [2,6,8,15,17].

GC is one of the most effective and precise methods for solving the problem of identifying FA components, since this method is multifaceted and highly sensitive, and has a small budget for analysis [18,19].

Various GC detectors, such as FPDs (flame photometric detectors), ECDs (electrolytic conductivity detectors), PIDs (photoionization detectors), FIDs (flame ionization detectors) and mass detectors (MS) have high sensitivity for detecting FAs. All of these detectors use nitrogen, helium and hydrogen as the carrier gas. However, an FID is widely used as a detector for detecting FAs. FIDs offer more sensitivity and a higher resolution than other detectors, and they are also reliable and relatively easy to use. In addition, various compositions can be detected using FIDs. These methods have been optimized for the analysis of specific foods, such as VOs [17,20–23].

Most studies regarding the use of GC for the analysis of polyunsaturated FAs, such as linoleic and linolenic acids and their isomers present in VOs, also recommend a high polarity column of the CP Sil 88 type. The main advantage is that this column can separate the isomers of linoleic and linolenic acids well [24–26].

A method for validating the determination of FAs is the procedure for the derivatization of FAs after the extraction of lipids from food products. Methods of FA derivatization



include acid or alkaline catalysis. Such methods of derivatization simplify the volatility of FAs, and also increase the separation and sensitivity of GC detectors [27,28].

Usually, the most common way to determine FAs is through lipid extraction, followed by the transformation of the FAs into methyl esters (methylation). In GC analysis, the identification of FAs is based on changing the FAs to the corresponding fatty acid methyl esters (FAMEs), which have a higher volatility [6,24]. The most commonly used base reagents for rapidly changing FAs into FAMEs are NaOH or KOH in methanol [29,30]. Other methods of methylation used benzyl alcohol, diazomethane in a mixture of ether, trichloride boron and sodium methoxide in methanol [31–34].

The purpose of the present work was to examine the authenticity of the main seven types of VOs produced by Kazakhstan's oilseed producers by identifying the indicators (FA composition) using gas chromatography with flame ionization detection. The main VOs established the accurate data of FAs that are available for scientists, as well as the food and energy industries. A multivariate statistical analysis was used to differentiate each of the vegetable oils.

## 2. Materials and Methods

### 2.1. Samples

In total, 35 VO samples were purchased from supermarkets in Almaty, Kazakhstan, which were produced by local manufacturers: sunflower oil ($n$ = 9), linseed oil ($n$ = 7), maize oil ($n$ = 4), safflower oil ($n$ = 3), cottonseed oil ($n$ = 3), soybean oil ($n$ = 6) and rapeseed oil ($n$ = 3). These represented the whole range of commercial trademarks available on the local market. Kazakhstan's markets sell many trademarked varieties of sunflower and linseed oils.

### 2.2. Preparation of Fatty Acid Methyl Esters (FAMEs) and GC/FID Analysis

The FAMEs were determined using a Shimadzu GC-2010 Plus gas chromatograph equipped with a flame ionization detection unit (Shimadzu, Kyoto, Japan). A high-polarity CP-Sil 2560 (100 m × 0.250 mm × 0.20 μm, Agilent Technologies, Santa Clara, CA, USA) fused silica column was used for chromatographic determination of the FAMEs. The carrier gas was nitrogen (99% purity), produced by a nitrogen generator (Parker Domnick Hunter G1110E, Hauppauge, NY, USA). The flow rate of hydrogen was 30 mL/min, the air flow rate was 300 mL/min, and the remaining flow was 30 mL/min. The parameters of gas chromatography were as follows: injector temperature, 250 °C; detector temperature, 260 °C; split mode, 1:40; and total flow, 95.5 mL/min. The initial temperature program of the column started from 100 °C for 5 min, then increased by 4 °C/min to 210 °C and held for 8 min, and then increased again by 10 °C/min to 240 °C and held for 16.5 min. The injection volume was 1.0 μL. Total time of analysis was 60 min. The analytical standard (37-component FAME Mix, Supelco, Merck, Darmstadt, Germany) was used to determine the FAMEs, then each identified compound was determined by normalization of the peak area (out of the sum of all peaks, we found the percentage of each compound) [35].

Next, 2.70 ± 0.01 g of sodium methylate powder (Sigma-Aldrich, St. Louis, MO, USA) was dissolved with 25 mL of absolute methanol (Sigma Aldrich, St. Louis, MO, USA) in a 50 mL volumetric flask. The solution was mixed and cooled to ambient temperature, and then 0.10 ± 0.01 mL of the oil was weighed in a 15 mL Falcon tube, to which 2 mL of n-hexane was added. Then, 0.1 mL of a sodium methylate solution in methanol was added and vortexed for 1 min. After the methylation reaction mixture had settled for 5 min and centrifuged at 3000 rpm for 5 min, 1 mL of the supernatant was transferred to a vial and injected for GC analysis [33,34].

### 2.3. Validation of the GC/FID Analysis

The analysis of the FAs using the GC/FID method was validated according to the guidelines of the ICH [36]. The method was validated in terms of linearity and the range of the FA calibration curves. Linearity was validated using the 37-component FAME Mix stan-

dard. Every FAME component was identified by the retention times and chromatograms in standard mix. The GC conditions were optimized to obtain a better separation of the FAs, such as the column temperature, flow rate and split ratio. The precision of the method was calculated by the repeatability. The precision was validated by repeating the process five times with a solution of the standard mix. The precision of the chromatographic system was confirmed by checking the %RSD of the retention times and peak areas. Five injections were performed over three days.

### 2.4. Statistical Analysis

The FA composition (relative percentage) was submitted to HCA and was performed using the Euclidean distances and Ward's method. Pearson's correlation coefficients were used to examine the relationships among the variables. PCA and ANN analyses based on FA components were used to obtain the mean value of oil. The values of each FA were compared by analysis of variance (ANOVA). When significant differences ($p < 0.05$) between the mean values were found, Tukey's test was performed. The least square means (LS means) test was applied to the results. All statistical analyses were carried out using JMP (JMP Statistical Discovery LLC, Cary, NC, USA) and Statistica 7 (StatSoft TIBCO Sofware Inc., Palo Alto, CA, USA).

## 3. Results and Discussion

### 3.1. Data Collection

In Kazakhstan, the main VOs used for consumption are sunflower, rapeseed, soybean and cottonseed oils, among which, the share of sunflower oil is more than 70% [14] (Figure 1). In 2020, 484.8 thousand tons of oil was produced in Kazakhstan, which was 46% more than the amount produced in 2015 (Table 1). The percentage share of sunflower oil in the period from 2015 to 2020 decreased from 77% to 70%, although the production of sunflower oil increased by more than 100 thousand tons during this period. All types of VOs are grown at a rate of 5–9% annually, and linseed and safflower oils are produced at the high rates in terms of percentage. Nevertheless, the production of VOs by Kazakhstan in 2020 was only 0.2% (0.48 million tons) of the total global volume [14,37].

**Table 1.** The total volume of VOs produced in Kazakhstan, according to the Bureau of National Statistics of the Republic of Kazakhstan [14].

| Type of Oil | Total Volume Produced in Kazakhstan, Thousand Tons | |
|---|---|---|
| | 2015 | 2020 |
| Cottonseed | 11.6 | 10.6 |
| Rapeseed | 31.3 | 68.8 |
| Soybean | 12.3 | 23.2 |
| Sunflower | 217.3 | 320.1 |
| Safflower | 2.3 | 6.0 |
| Linseed | 6.2 | 29.1 |
| Maize | 0.04 | 0.3 |
| Total | 225.7 | 484.8 |

### 3.2. Validation of the Method

The method of determining the FAs was validated using the 37-component FAME Mix standard. All FA components were identified by the retention times and chromatograms in the standard mix standards (Table 2, Figure 2).

A quantitative analysis of the FAs was carried out and calibration curves were plotted between 20.2 µg/mL and 612 µg/mL. The calibration curves consisted of five concentrations. Table 2 showed the retention times, correlation coefficients, linear equations, ranges, limits of detection (LOD) and limits of quantification (LOQ) for each FA component. The correlation coefficients were more than 98%, confirming the detector's excellent response.

The accuracy and precision of the method were determined by a replicate analysis of the FAME Mix standard. The repeatability for the standards, in terms of the calculated %RSD for the retention times, was not greater than 0.5%. When we calculated the peak areas that were not greater than 1.0% under the same conditions, the precision of the retention times was not greater than 0.3%. The LOD value was between 0.29 µg/mL and 1.95 µg/mL, and the LOQ values varied between 2.06 µg/mL and 3.95 µg/mL, which showed that the method is sensitive. According to the data determined above, this method could be used for the identification of FAs in VO products, conforming with the ICH guidelines [36].

**Table 2.** FAME retention times, LOD and LOQ values, and linear parameters for the FA standards from calibration curves.

| # | Fatty Acid Components | RT (Mean) | $R^2$ | Calibration Curve Equation | Range (µg/mL) | LOD (µg/mL) | LOQ (µg/mL) |
|---|---|---|---|---|---|---|---|
| 1 | C4:0 | 7.086 | 0.9972 | $y = 0.0039x + 29.719$ | 40.4–404 | 1.22 | 3.73 |
| 2 | C6:0 | 8.800 | 0.9958 | $y = 0.0033x + 23.897$ | 40.4–404 | 1.73 | 3.27 |
| 3 | C8:0 | 11.905 | 0.9996 | $y = 0.0036x + 21.439$ | 40.4–404 | 0.88 | 2.67 |
| 4 | C10:0 | 15.971 | 0.9998 | $y = 0.0038x + 1.8014$ | 40.8–408 | 0.60 | 2.81 |
| 5 | C11:0 | 18.356 | 0.9998 | $y = 0.0039x - 0.1529$ | 20.4–204 | 0.45 | 2.36 |
| 6 | C12:0 | 20.112 | 0.9985 | $y = 0.0033x + 14.09$ | 40.4–404 | 0.82 | 2.50 |
| 7 | C13:0 | 22.28 | 0.9991 | $y = 0.0033x + 1.6595$ | 20.3–203 | 0.87 | 2.65 |
| 8 | C14:0 | 23.83 | 0.9985 | $y = 0.0029x + 15.744$ | 40.4–404 | 1.63 | 3.95 |
| 9 | C14:1 | 25.412 | 0.9880 | $y = 0.004x + 12.928$ | 20.4–204 | 1.29 | 2.97 |
| 10 | C15:0 | 25.731 | 0.9997 | $y = 0.0024x + 0.1732$ | 20.3–203 | 0.54 | 2.64 |
| 11 | C15:1 | 27.043 | 0.9997 | $y = 0.0039x + 15.084$ | 20.4–204 | 0.48 | 2.46 |
| 12 | C16:0 | 27.214 | 0.9999 | $y = 0.0026x - 3.5892$ | 61.2–612 | 1.49 | 2.48 |
| 13 | C16:1 | 28.597 | 0.9971 | $y = 0.0031x + 10.884$ | 20.4–204 | 1.60 | 2.87 |
| 14 | C17:0 | 29.07 | 0.9999 | $y = 0.0027x + 1.9941$ | 21.0–210 | 0.26 | 2.79 |
| 15 | C17:1 | 30.136 | 0.9999 | $y = 0.003x + 8.7811$ | 20.4–204 | 0.31 | 2.95 |
| 16 | C18:0 | 30.464 | 0.9999 | $y = 0.0025x + 10.246$ | 40.8–408 | 0.44 | 2.34 |
| 17 | C18:1n9t | 31.149 | 0.9978 | $y = 0.003x + 16.506$ | 20.2–202 | 1.39 | 3.22 |
| 18 | C18:1n9c | 31.403 | 0.9999 | $y = 0.0022x + 22.074$ | 40.4–404 | 0.47 | 2.43 |
| 19 | C18:2n6t | 32.306 | 0.9903 | $y = 0.0025x + 14.645$ | 20.2–202 | 1.95 | 2.94 |
| 20 | C18:2n6c | 32.993 | 0.9951 | $y = 0.0024x + 20.107$ | 20.2–202 | 1.11 | 3.40 |
| 21 | C20:0 | 33.491 | 0.9999 | $y = 0.0025x + 18.09$ | 40.8–408 | 0.35 | 2.06 |
| 22 | C18:3n6c | 34.002 | 0.9999 | $y = 0.0027x + 17.388$ | 20.3–203 | 0.32 | 2.98 |
| 23 | C20:1n9c | 34.512 | 0.9933 | $y = 0.0043x + 8.5154$ | 20.2–202 | 1.44 | 2.41 |
| 24 | C18:3n3c | 34.693 | 0.9993 | $y = 0.0021x + 20.688$ | 20.4–204 | 0.77 | 2.35 |
| 25 | C21:0 | 35.159 | 0.9986 | $y = 0.0022x + 10.468$ | 20.3–203 | 1.12 | 3.41 |
| 26 | C20:2 | 36.134 | 0.9983 | $y = 0.0033x + 4.2141$ | 20.4–204 | 1.25 | 3.80 |
| 27 | C22:0 | 36.835 | 0.9975 | $y = 0.0026x + 9.5838$ | 40.5–405 | 1.11 | 2.40 |
| 28 | C20:3 | 37.412 | 0.9948 | $y = 0.0031x + 6.2057$ | 20.4–204 | 1.16 | 2.56 |
| 29 | C22:1 | 38.04 | 0.9999 | $y = 0.0059x + 12.565$ | 20.4–204 | 0.29 | 2.88 |
| 30 | C20:3 | 38.175 | 0.9997 | $y = 0.0023x + 18.341$ | 20.4–204 | 0.56 | 2.70 |
| 31 | C23:0 | 38.455 | 0.9931 | $y = 0.0027x + 14.205$ | 20.3–203 | 1.48 | 2.53 |
| 32 | C20:4 | 38.903 | 0.9947 | $y = 0.0022x + 11.377$ | 20.2–202 | 1.17 | 2.60 |
| 33 | C22:2 | 40.02 | 0.9981 | $y = 0.0035x + 14.392$ | 20.4–204 | 1.29 | 2.91 |
| 34 | C24:0 | 41.09 | 0.9998 | $y = 0.0023x + 39.734$ | 40.4–404 | 0.58 | 2.75 |
| 35 | C20:5 | 41.403 | 0.9994 | $y = 0.0091x + 29.666$ | 20.4–204 | 0.75 | 2.28 |
| 36 | C24:1 | 42.207 | 0.9961 | $y = 0.0026x + 41.385$ | 20.4–204 | 1.58 | 2.79 |
| 37 | C22:6 | 45.503 | 0.9995 | $y = 0.0087x + 34.921$ | 20.3–203 | 0.55 | 2.69 |

RT, retention time; R, correlation coefficient; LOD, limit of detection; LOQ, limit of quantification.

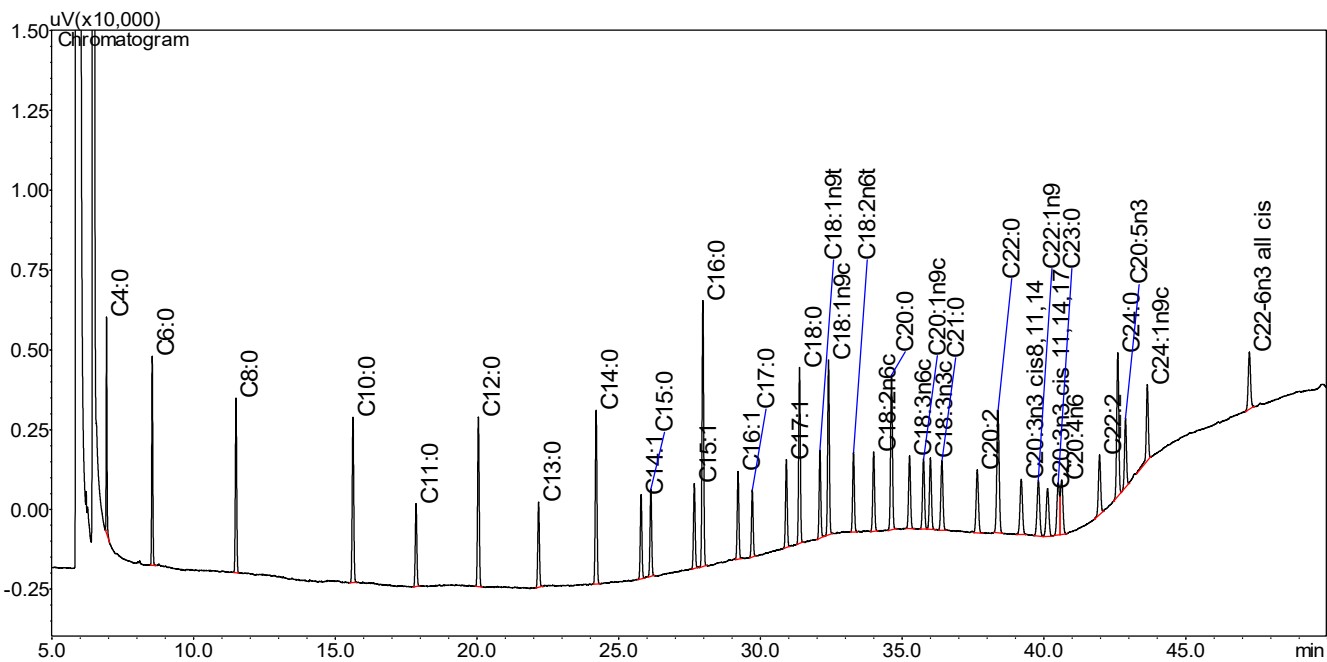

**Figure 2.** GC chromatogram of the 37-component FAME Mix standard analyzed on a high-polarity capillary column CP-Sil 2560. The column's temperature was programmed as follows: 100 °C (5 min); 40 °C/min to 210 °C (8 min); 10 °C/min to 240 °C (16.5 min).

### 3.3. Determination of FAs in Vegetable Oils

The FA profiles of seven commercially available VO types were determined (Table 3). All the oils' FA compositions were analyzed in duplicate and methylated on the same day. Moreover, all samples were analyzed only on the CP-Sil 2560 capillary column (100 m × 0.250 mm × 0.20 µm). Using a 100 m high-polarity capillary column allowed a good separation of many FAs, as well as *cis-* and *trans-*isomers [24,25,38]. The oils analyzed via GC chromatogram are shown in Figure S1. The composition of all FAs complied with the international food standards of the Codex Alimentarius Codex Standard 210 "Standard for named vegetable oils" [39].

**Table 3.** FA composition and main specific ratios of the seven VOs.

| Fatty Acid | | FA Content, %, Mean ± SD | | | | | | |
|---|---|---|---|---|---|---|---|---|
| | | Safflower (*n* = 3) | Sunflower (*n* = 9) | Maize (*n* = 4) | Linseed (*n* = 7) | Cottonseed (*n* = 3) | Soybean (*n* = 6) | Rapeseed (*n* = 3) |
| C14:0 | Myristic | ND | ND | ND | ND | 0.68 ± 0.05 [d] | ND | ND |
| C16:0 | Palmitic | 6.90 ± 0.24 [c] | 7.24 ± 0.18 [c] | 8.33 ± 1.27 [c] | 5.84 ± 0.24 [c] | 21.66 ± 0.58 [b] | 11.45 ± 0.51 [c] | 4.37 ± 0.8 [c] |
| C16:1 | Palmitoleic | 0.09 ± 0.01 [d] | 0.06 ± 0.01 [d] | 0.05 ± 0.01 [d] | 0.08 ± 0.00 [d] | 0.46 ± 0.03 [d] | ND | ND |
| C18:0 | Stearic | 2.50 ± 0.16 [c] | 5.26 ± 0.75 [c] | 3.72 ± 0.78 [c] | 4.26 ± 0.07 [c] | 2.98 ± 0.22 [c] | 6.34 ± 0.81 [c] | 2.60 ± 0.34 [c] |
| C18:1n9t | Oleic trans | ND | ND | ND | ND | 0.30 ± 0.01 [d] | ND | ND |
| C18:1n9c | Oleic | 15.50 ± 0.28 [c] | 18.46 ± 0.53 [bc] | 23.36 ± 4.37 [bc] | 17.52 ± 0.49 [c] | 16.82 ± 0.21 [c] | 25.35 ± 0.64 [bc] | 66.31 ± 2.03 [a] |
| C18:2n6t | Linoleic trans | 0.07 ± 0.01 [d] | 1.26 ± 0.02 [d] | 0.69 ± 0.55 [d] | 0.04 ± 0.00 [d] | 0.22 ± 0.01 [d] | ND | ND |
| C18:2n6c | Linoleic | 74.25 ± 0.63 [a] | 66.32 ± 0.64 [a] | 62.36 ± 4.60 [a] | 15.54 ± 0.15 [c] | 55.93 ± 0.01 [a] | 50.91 ± 1.34 [a] | 18.62 ± 1.3 [bc] |
| C20:0 | Arachidic | 0.23 ± 0.02 [d] | 0.23 ± 0.04 [d] | 0.32 ± 0.13 [d] | 0.11 ± 0.00 [d] | 0.25 ± 0.01 [d] | 0.21 ± 0.03 [d] | 0.31 ± 0.06 [d] |
| C20:1n9c | Gondoic | 0.08 ± 0.02 [d] | ND | 0.19 ± 0.05 [d] | ND | 0.13 ± 0.01 [d] | ND | ND |
| C18:3n6c | γ-Linolenic | ND | 0.11 ± 0.01 [d] | ND | 0.18 ± 0.00 [d] | ND | ND | ND |
| C18:3n3c | α-Linolenic | 0.07 ± 0.02 [d] | 0.09 ± 0.01 [d] | 0.55 ± 0.12 [d] | 56.01 ± 0.33 [a] | 0.05 ± 0.03 [d] | 5.24 ± 0.35 [cd] | 6.93 ± 1.33 [c] |

**Table 3.** *Cont.*

| Fatty Acid | | FA Content, %, Mean ± SD | | | | | | |
|---|---|---|---|---|---|---|---|---|
| | | Safflower (*n* = 3) | Sunflower (*n* = 9) | Maize (*n* = 4) | Linseed (*n* = 7) | Cottonseed (*n* = 3) | Soybean (*n* = 6) | Rapeseed (*n* = 3) |
| C20:2 | Eicosadienoic | ND | 0.34 ± 0.29 [d] | 0.05 ± 0.01 [d] | ND | 0.12 ± 0.02 [d] | ND | ND |
| C22:2 | Docosadienoic | ND | 0.16 ± 0.01 [d] | ND | ND | ND | ND | ND |
| C22:0 | Behenic | 0.12 ± 0.02 [d] | 0.43 ± 0.02 [d] | 0.37 ± 0.08 [d] | ND | ND | 0.26 ± 0.08 [d] | ND |
| C24:0 | Lignoceric | 0.07 ± 0.01 [d] | 0.11 ± 0.01 [d] | 0.14 ± 0.04 [d] | ND | ND | ND | ND |
| C22:1 | Erucic | ND | ND | ND | ND | ND | ND | 0.1 ± 0.02 [d] |
| C22-6n3 | Docosahexaenoic | 0.07 ± 0.01 [d] | 0.06 ± 0.01 [d] | 0.05 ± 0.01 [d] | ND | 0.22 ± 0.05 [d] | ND | ND |
| SFAs | | 9.81 ± 0.44 | 13.27 ± 0.99 | 12.89 ± 2.25 | 10.20 ± 0.31 | 25.57 ± 0.81 | 18.28 ± 1.44 | 7.3 ± 1.2 |
| USFAs | | 90.06 ± 0.21 | 85.11 ± 1.52 | 86.60 ± 4.57 | 89.16 ± 0.48 | 73.69 ± 0.31 | 81.52 ± 1.17 | 91.88 ± 2.34 |
| MUFAs | | 15.67 ± 0.31 | 18.52 ± 0.54 | 23.60 ± 4.42 | 17.61 ± 0.49 | 17.42 ± 0.25 | 25.36 ± 0.65 | 66.32 ± 2.04 |
| PUFAs | | 74.39 ± 0.09 | 66.59 ± 0.94 | 63.00 ± 4.72 | 71.55 ± 0.48 | 56.27 ± 0.06 | 56.16 ± 1.71 | 25.56 ± 2.65 |
| Omega-6 | | 74.25 ± 0.66 | 66.43 ± 0.64 | 62.36 ± 4.60 | 15.72 ± 0.15 | 55.93 ± 0.01 | 50.91 ± 1.35 | 18.63 ± 1.31 |
| Omega-3 | | 0.14 ± 0.02 | 0.15 ± 0.01 | 0.60 ± 0.12 | 56.01 ± 0.33 | 0.27 ± 0.03 | 5.24 ± 0.36 | 6.94 ± 1.34 |
| PUFA/SFA | | 7.58 | 5.02 | 4.89 | 7.01 | 2.20 | 3.07 | 3.50 |
| Omega-6/Omega-3 | | 530.35 | 442.86 | 103.93 | 0.28 | 207.14 | 9.70 | 2.68 |

All values are expressed as the mean ± standard deviation (SD). ND, not detected; SFAs, saturated fatty acids; USFAs, unsaturated fatty acids; MUFAs, monounsaturated fatty acids; PUFAs, polyunsaturated fatty acids. Values within each column followed by different letters are significantly different (*p* < 0.05).

These were grouped into saturated (SFAs), monounsaturated (MUFAs), and polyunsaturated (PUFAs) fatty acids, and their respective calculation formulae were:

$$\text{SFAs} = \text{C4:0} + \text{C6:0} + \text{C8:0} + \text{C10:0} + \text{C11:0} + \text{C12:0} + \text{C13:0} + \text{C14:0} + \text{C15:0} + \text{C16:0} + \text{C17:0} + \text{C18:0} + \text{C20:0} + \text{C21:0} + \text{C22:0} + \text{C23:0} + \text{C24:0} \tag{1}$$

$$\text{MUFAs} = \text{C14:1} + \text{C15:1} + \text{C16:1} + \text{C17:1} + \text{C18:1n9c} + \text{C20:1n9c} + \text{C22:1} + \text{C24:1} \tag{2}$$

$$\text{PUFAs} = \text{C18:2n6c} + \text{C18:3n6c} + \text{C18:3n3c} + \text{C20:2} + \text{C20:3} + \text{C20:4} + \text{C22:2} + \text{C20:5} + \text{C22:6} \tag{3}$$

Figure 3 shows the SFA, MUFA and PUFA compounds of the different oils analyzed. In all oils, palmitic acid C16:0 was always the main SFA, where the percentages were between 5.59% and 22.24%, followed by stearic acid C18:0 (2.34–6.34%). The content of other SFAs, such as C14:0, C20:0, C22:0 and C24:0, was too small, being less than 1% in total. Both MUFA and PUFA compounds showed variations depending on the VOs and varied from 15.67% to 66.32% (MUFAs) and from 25.56% to 74.39% (PUFAs) of the total FAs. Oleic acid C18:1n9c was always the main MUFA, with a content between 15.78% and 66.31%. Rapeseed oil had C18:1n9c (a MUFA) as its main oleic acid (66.31%). Except in linseed oil, linoleic acid C18:2n6c was the major PUFA in all of the VOs (18.62–74.19%). Linseed oil was predominantly the PUFA α-linolenic acid C18:3n3c (55.67%). Cottonseed oil had the highest SFA content (25.8%), of which palmitic acid C16:0 was the main SFA (21.66%). Soybean oil had the highest content of stearic acid (the second main SFA) at 6.34%.

From the data shown in Figure 3, it was found that SFAs predominated in cottonseed oil, and USFAs predominated in safflower oil. Therefore, safflower is a valuable oil-containing crop, and the oil does not belong to the VO with high oleic acid. Rapeseed oils had a higher proportion of MUFAs than the other VOs researched. The ratio of the content of PUFAs to the content of SFAs: in safflower oil was 7.58%; in sunflower oil was 5.02%; in maize oil was 4.89%; in linseed oil was 7.01%; in cottonseed oil was 2.20%; in soybean oil was 3.07%; and in rapeseed oil was 3.50%. These levels were optimal for these types of VOs, thus indicating their high nutritional quality. The values of the PUFA/SFA ratios are recommended to be greater than 0.45 [40,41].

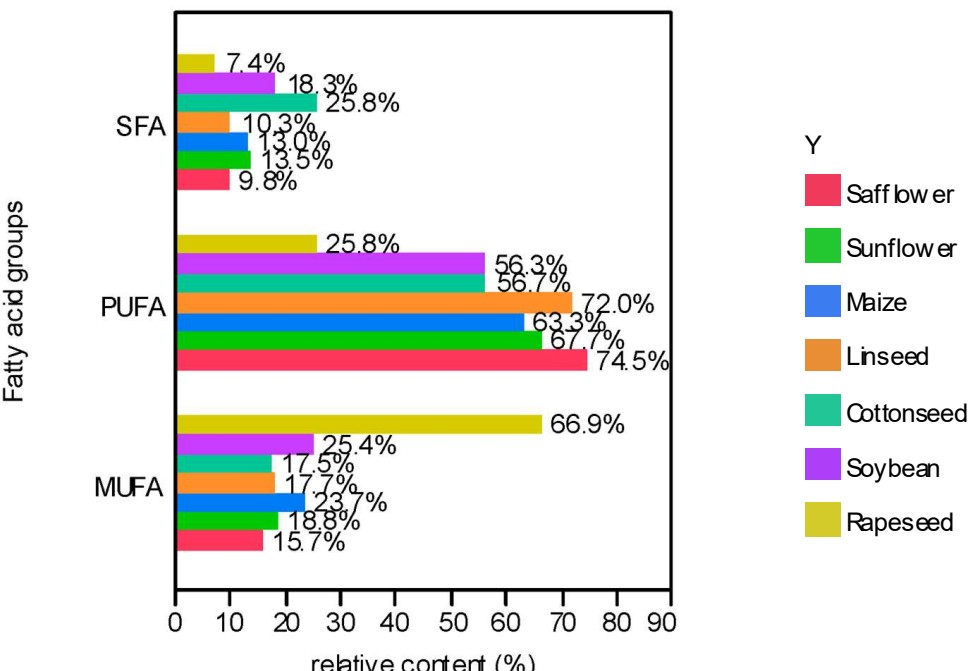

**Figure 3.** Average values of the SFA, MUFA and PUFA groups (percentage of total FA) in the vegetable oils.

According to the data given in Table 3, safflower oil had the greatest mass fraction of linoleic acid C18:2n6c (74.25%). Some reports have shown [42,43] that the FA composition can depend on the influence of the environmental conditions. The samples did not include oils containing FAs with a low molecular weight, such as capronic acid C6:0, caprylic acid C8:0, capric acid C10:0 and lauric acid C12:0. The content of palmitic acid C16:0 was 6.90%. The content of stearic acid C18:0 was around 2.50%. In the samples, the content of oleic acid C18:1n9c was 15.50%, that of linolenic acid C18:3n3c was 0.07% and that of arachidic acid C20:0 was 0.23%. The mass fractions of the following FAs were represented in amounts not exceeding 0.5%: behenic acid C22:0 and lignoceric acid C24:0. Similar results have previously been reported [43]. Moreover, the sowing time also affected oleic and linolenic acids to some degree. A high content of the total amount of USFAs was found in safflower (90.06%), sunflower (85.11%) and linseed oils (89.16%), the results of which corresponded to the research information [44]. Sabzalian et al. [45] reported that FA compositions in cultivated and wild species of safflower oil were not considerably different. Mihaela et al. [46] reported a higher ratio of SFAs (17.1%) than Giakoumis' [47] and Kazakhstan's safflower oil, at 10.1 and 9.8, respectively, as shown in Table 3 and Figure S5.

According to Table 3, the composition of sunflower oil's FAs showed an absence of some FAs, namely myristic acid C14:0 and gondoic acid C20:1. The content of palmitoleic acid C16:1 was in a range of up to 0.3%. Saturated stearic acid C18:0 did not exceed the limit of 6.5% (5.26%). Linoleic acid C18:2n6c (66.32%) was dominant in the composition of triglycerides of the sunflower oil sample. Oleic acid C18:1 was present in an amount of 18.46%. The content of palmitic acid C16:0 was 7.24%. Sunflower oil's FA contents were the same as the Codex Stan values [39]. Similar to safflower oil, sunflower oil's main FA components significantly depend on the growing area [48] and the refining processes [49]. The content of linolenic acid C18:3n6c was 0.17%, and the contents of the SFA arachidic acid C20:0, behenic acid C22:0 and lignoceric acid C24:0 were 0.23%, 0.44% and 0.11%, respectively. Many sources reported [47,50–52] the same results of SFA variations (10.7–13.3%). But, Vingering et al. [50] and Özogul et al. [52] showed lower PUFA ratios, at 57.4 and 59.5%, respectively, which are similar to other sources' results [47,51] as well as to the Kazakhstani sunflower oil results, 63.5–67.6%, respectively (Table 4, Figure S5).

**Table 4.** Comparative study of FA composition of various VOs from different sources.

| Oil Type | Data Sources | Fatty Acids Contents, % | | | | | | | | | | | |
|---|---|---|---|---|---|---|---|---|---|---|---|---|---|
| | | C14:0 | C16:0 | C16:1 | C18:0 | C18:1 | C18:2 | C18:3 | C20:0 | C20:1 | C22:0 | C22:1 | C24:0 |
| Safflower oil | Kazakhstan's | - | 6.90 | 0.09 | 2.50 | 15.50 | 74.25 | 0.07 | 0.23 | 0.08 | 0.12 | - | 0.07 |
| | Giakoumis [47] | 0.12 | 7.41 | 0.04 | 2.36 | 14.37 | 75.17 | 0.05 | 0.08 | - | 0.10 | - | - |
| | Mihaela et al. [46] | - | 11.07 | - | 4.37 | 12.76 | 69.65 | 0.49 | 0.78 | - | 0.59 | - | 0.29 |
| Sunflower oil | Kazakhstan's | - | 7.24 | 0.06 | 5.26 | 18.46 | 66.32 | 0.09 | 0.23 | - | 0.43 | - | 0.11 |
| | Kim et al. [51] | - | 5.83 | - | 3.24 | 26.28 | 62.97 | 0.38 | 0.21 | 0.15 | 0.59 | - | 0.19 |
| | Vingering et al. [50] | 0.1 | 6.0 | 0.1 | 3.6 | 29.4 | 54.5 | 0.1 | 0.3 | 0.2 | 0.7 | - | 0.2 |
| | Giakoumis [47] | 0.04 | 6.35 | 0.07 | 3.92 | 20.91 | 67.58 | 0.17 | 0.22 | 0.11 | 0.66 | - | 0.26 |
| | Özogul et al. [52] | - | 5.99 | 0.27 | 4.57 | 29.14 | 58.58 | 0.07 | - | - | - | - | - |
| Maize oil | Kazakhstan's | - | 8.33 | 0.05 | 3.72 | 23.36 | 62.36 | 0.55 | 0.32 | 0.19 | 0.37 | - | 0.14 |
| | Kim et al. [51] | - | 10.85 | - | 1.74 | 29.48 | 55.80 | 1.07 | 0.41 | 0.20 | 0.13 | - | 0.17 |
| | Giakoumis [47] | - | 11.88 | 0.13 | 2.10 | 27.23 | 57.74 | 0.64 | 0.32 | 0.35 | - | - | 0.14 |
| | Özogul et al. [52] | - | 10.19 | 0.50 | 2.62 | 32.23 | 52.91 | 0.85 | - | - | - | - | - |
| Linseed oil | Kazakhstan's | - | 5.84 | 0.08 | 4.26 | 17.52 | 15.54 | 56.01 | 0.11 | - | - | - | - |
| | Kim et al. [51] | - | 4.54 | - | 3.32 | 18.20 | 16.60 | 56.66 | 0.12 | 0.12 | 0.11 | - | - |
| | Giakoumis [47] | 0.04 | 5.18 | 0.10 | 3.26 | 19.04 | 16.12 | 54.59 | 0.09 | 0.07 | 0.10 | 0.20 | 0.03 |
| Cottonseed oil | Kazakhstan's | 0.68 | 21.66 | 0.46 | 2.98 | 16.82 | 55.93 | 0.05 | 0.25 | 0.13 | - | - | - |
| | Giakoumis [47] | 0.72 | 25.19 | 0.36 | 1.79 | 16.47 | 54.83 | 0.19 | 0.22 | 0.07 | 0.11 | - | - |
| Soybean oil | Kazakhstan's | - | 11.45 | - | 6.34 | 25.35 | 50.91 | 5.24 | 0.21 | - | 0.26 | - | - |
| | Kim et al. [51] | - | 10.10 | - | 3.94 | 22.47 | 55.17 | 6.51 | 0.31 | 0.27 | 0.34 | - | 0.11 |
| | Giakoumis [47] | 0.12 | 11.50 | 0.16 | 4.11 | 23.50 | 53.33 | 6.76 | 0.32 | 0.22 | 0.27 | 0.07 | 0.13 |
| Rapeseed oil | Kazakhstan's | - | 4.37 | - | 2.60 | 66.31 | 18.62 | 6.93 | 0.31 | - | - | 0.1 | - |
| | Vingering et al. [50] | 0.1 | 4.5 | 0.2 | 1.6 | 55.2 | 19.4 | 7.8 | 0.6 | 1.1 | 0.3 | 0.2 | - |
| | Giakoumis [47] | 0.04 | 4.06 | 0.23 | 1.54 | 62.29 | 20.65 | 8.71 | 0.87 | 1.09 | 0.27 | 0.77 | 0.04 |

The chemical composition of maize oil was similar to that of sunflower oil [53]. According to Table 3, there was no lauric acid C12:0 or myristic acid C14:0 in the maize oil sample. Some sources have reported the presence of medium-chain fatty acids, such as C12:0 and C14:0 [54]. The FA components of maize oil were close to the Codex Stan [39] requirements but had slight deviations in the content of oleic acid C18:1 (23.36%, normally 20.0–42.0%) and linoleic acid C18:2n6c (62.36%, normally 34.0–65.0%). Our analysis of maize oil's FAs showed a significantly higher value for PUFAs [55], namely 59%. The SFA palmitic acid C16:0 and stearic acid C18:0 had values of 8.33% and 3.72%, respectively. Özogul et al. [52] reported a higher percentage of MUFAs (33.0%) that was significantly different than Kazakhstani oil producers, where MUFAs consisted only 23.7%.

The samples indicated the most important components of linseed oil, which caused its high biological activity: linolenic acid C18:3n6c, 56.01%; oleic acid C18:1n9c, 17.52%; and linoleic acid C18:2n6c, 15.54%. The total content of PUFAs was 72%, which was in line with the major USFA values found by other authors [56,57]. The content of the SFAs were as follows: stearic acid C18:0, 4.26%; palmitic acid C16:0, 5.84%; and arachidic acid C20:0, 0.11%. Depending on the specific FA composition, linseed oil belongs to the group of oils with a linolenic acid content of more than 20%. The investigated FA contents had great differences in several factors, such as genetic factors, geographical site, growth factors, environmental conditions and processing conditions [58,59]. Kim et al. [51] and Giakoumis [47] reported the same results for SFAs (8.8–10.3%), MUFAs (17.7–19.6%) and PUFAs (71.6–73.5%) with Kazakhstan's linseed oil producer's results.

According to Table 3, the composition of cottonseed oil samples was characterized by the following composition of major FAs: linoleic acid C18:2n6c, 55.93%; oleic acid C18:1n9c,

16.82; and palmitic acid C16:0, 21.66%. Cottonseed oils had the highest SFA content in our research: 25.57%. Other authors' research found the same percentage of linoleic acid C18:2n6c, oleic acid C18:1n9c and palmitic acid C16:0 [60,61]. In addition to the main FAs, cottonseed oil also consists of minor FAs (0.1–1% each acid) such as myristic C14:0, palmitoleic C16:1, arachidic C20:0 and α-linolenic C18:3n3c acids [62].

The results for soybean oils showed a predominance of linoleic acid C18:2n6c (50.91%), oleic acid C18:1n9c (25.35%) and palmitic acid C16:0 (11.45%). Soybean oil had a considerably high steric acid content (6.34%) than the other VOs researched. Ivanov et al. [63] found 5.15%, and Li et al. [64] established a content of 4.80%. Fehr [65] successfully reported three modified soybean oil species, where the modification of the FA composition of linoleic acid content reduced and eliminated the hydrogenation process. This process made it possible to increase oleic acid from 25% to 80%. Kim et al. [51] reported a higher PUFA percentage (62.2%) that was almost 6% more than that of Kazakhstani oil producers.

Rapeseed is a MUFA-rich oil that is especially rich in oleic acid C18:1n9c (66.31%). This was the highest oleic acid content in our research and this result was higher than that found in other research [50,64], namely 60.92% and 55.2%, respectively, for Ivanov et al. and Li et al. Alongside the high content of oleic acid as the main FAs, rapeseed oil had higher contents of α-linolenic C18:3n3 acid (6.94%) and linoleic acid C18:2n6c (18.62%). Rapeseed oil contained erucic acid C22:1 (0.1%); this oil was the only one with erucic acid, which is typical of this type of oil [66].

### 3.4. Omega-6/Omega-3 Ratio in VOs

The omega-6/omega-3 ratio is considered to be an indicator for comparing the nutritional value of VOs. In our research, the lowest omega-6/omega-3 ratios were found for linseed oil (0.28%), rapeseed oil (2.68%) and soybean oil (9.70%). These ratios were similar to the data from the literature [50,61]. The high contents of α-linolenic acid C18:3n3 as the main FAs in linseed resulted in a low omega-6/omega-3 ratio. Because of the richer linoleic acid C18:2n6c and poorer α-linolenic acid C18:3n3 and docosahexaenoic acid C22:6n3, maize oil, cottonseed oil, sunflower oil and safflower oil showed higher omega-6/omega-3 ratios of 103.93, 207.14, 442.86 and 530.35, respectively. The results presented for maize and cottonseed oils had slightly different ratios. Nevertheless, sunflower and safflower oils had great differences from those published by Dubois et al. [61], namely ratios of 131 and 253 for sunflower and safflower, respectively.

### 3.5. Cluster Analysis

Hierarchical clustering analysis (HCA) using Ward's linkage was also carried out in order to determine the relationship between the VOs on the basis of their FA compounds. The dendrograms classified the species into two groups (Figure 4). The first group was characterized by higher linoleic and oleic acid contents. The second group, including linseed oil, was characterized by higher concentrations of linolenic acid. Safflower, sunflower, maize, cottonseed and soybean oils were determined as the most similar species with the highest linoleic acid content. Linseed oil had the highest level of linolenic acid; it was different among the studied species in terms of its FA composition. The proportion of α-linolenic acid C18:3n3 in the FA composition of linseed oil reached 56.01%, while two VOs (safflower and maize) had less than 1%, and another two VOs had none at all. Rapeseed oil had a higher oleic acid C18:1n9c content of 66.31%; at the same time, other VOs had contents lower than 25%.

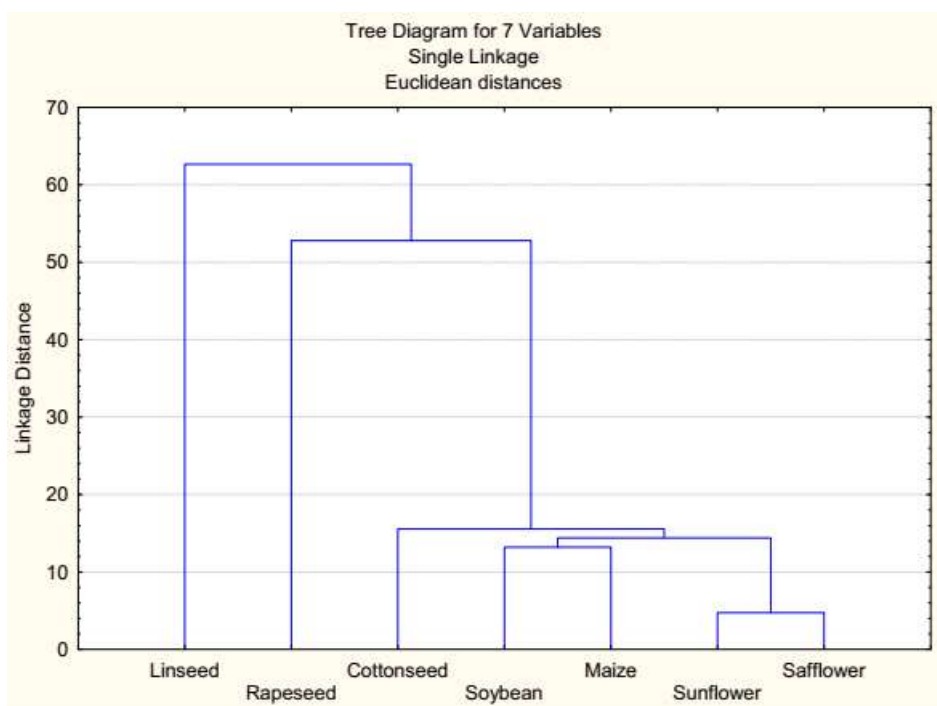

**Figure 4.** Dendrograms of the clustering analyses of seven VOs based on each oil's FA composition. The vertical distance shows the Euclidean distance.

The Euclidean distances of the seven FAME clusters were compared within each of the oil types, as presented in Table 5. As shown in Table 5, safflower, sunflower and maize oils were very closely related to each other, with Euclidean distances of 6.6–14.4. Cottonseed oil had a slight similarity to sunflower and maize oils, with an average distance between 16.2 and 18.1. Linseed oil was more distinct, with an Euclidean distance of 70.9–81.1, and rapeseed oils had a distance of 52.8–75.7 from other oils. CA was used to study the influence of FA composition on the clustering variation as well as other data processing parameters, such as sterols, tocopherol and oil stability index [64,67,68].

**Table 5.** Euclidean distances.

|  | Safflower | Sunflower | Maize | Linseed | Cottonseed | Soybean | Rapeseed |
|---|---|---|---|---|---|---|---|
| Safflower | 0.0 | 9.0 | 14.4 | 81.1 | 23.6 | 26.5 | 75.7 |
| Sunflower | 9.0 | 0.0 | 6.6 | 75.6 | 18.1 | 18.2 | 68.0 |
| Maize | 14.4 | 6.6 | 0.0 | 72.9 | 16.2 | 13.2 | 61.8 |
| Linseed | 81.1 | 75.6 | 72.9 | 0.0 | 70.9 | 62.7 | 69.3 |
| Cottonseed | 23.6 | 18.1 | 16.2 | 70.9 | 0.0 | 15.5 | 64.7 |
| Soybean | 26.5 | 18.2 | 13.2 | 62.7 | 15.5 | 0.0 | 52.8 |
| Rapeseed | 75.7 | 68.0 | 61.8 | 69.3 | 64.7 | 52.8 | 0.0 |

### 3.6. Correlation Study

Table 6 provided the Pearson correlation coefficients between the main FA parameters. Those correlations demonstrated significantly high positive correlations between PUFAs and USFAs ($r = 0.908$, $p < 0.033$), USFAs and PUFAs/SFAs ($r = 0.9502$, $p < 0.013$), and PUFAs and PUFAs/SFAs ($r = 0.9859$, $p < 0.001$), as shown in Table 6. SFAs had significant negative correlations with USFAs ($r = -0.9966$, $p < 0.001$), PUFAs ($r = -0.9010$, $p < 0.011$) and PUFAs/SFAs ($r = -0.9369$, $p < 0.018$). Omega-3 had a significantly negative correlation with omega-6 ($r = -0.9573$, $p < 0.024$) (Figure S2). Our results showed that SFAs had no positive correlation between all parameters, except omega-6 (0.10), which may have low concentrations of the MUFA ratio from most of the VOs. Omega-6/omega-3 presented a

very similar correlation to all of FA ratios. Strong positive and negative correlations were reported between FA groups [69,70].

**Table 6.** Pearson correlation coefficient (r) was between FA parameters.

|  | SFAs | USFA | MUFA | PUFA | Omega-6 | Omega-3 | PUFAs/SFAs | Omega-6/Omega-3 |
|---|---|---|---|---|---|---|---|---|
| SFAs | 1.0000 | −0.9966 | −0.0393 | −0.9010 | 0.1063 | −0.3616 | −0.9369 | −0.4159 |
| USFAs | *** | 1.0000 | 0.0301 | 0.9080 | −0.1053 | 0.3626 | 0.9502 | 0.4622 |
| MUFAs | * | * | 1.0000 | −0.3915 | 0.0628 | −0.1698 | −0.2650 | −0.4448 |
| PUFAs | ** | ** | * | 1.0000 | −0.1233 | 0.4050 | 0.9859 | 0.6120 |
| Omega-6 | * | * | * | * | 1.0000 | −0.9573 | −0.1666 | 0.5022 |
| Omega-3 | * | * | * | * | ** | 1.0000 | 0.4412 | −0.2847 |
| PUFAs/SFAs | ** | ** | * | *** | * | * | 1.0000 | 0.5906 |
| Omega-6/ Omega-3 | * | * | * | * | * | * | * | 1.0000 |

The upper triangle shows the Pearson correlation coefficients, and the lower triangle shows the significance levels (*, less than 0.05; **, less than 0.01; ***, less than 0.001).

### 3.7. Principal Component Analysis

In our research, PCA was used as a multidimensional statistical analysis for the possible identification of VOs on dependent variables—FA composition. FA compositions serve as the main component. In Figure 5, PCA explained 90% of total variation with PC1 for 75% and PC2 for 15%. The PCA plot in Figure 5 shows that seven different VOs formed a distinct group, and each group could be distinguished. The group consisting of safflower, sunflower, maize, cottonseed and soybean oils were close together due to their similar FA compositions. Another group of oils comprised rapeseed and linseed oils. According to Figure 5, the principal scores among VOs were mainly related to linoleic acid C18:2n6c, where safflower, sunflower, maize, cottonseed and soybean oils possessed higher concentrations of linoleic acid. As shown in Figure 5, the group of linseed and rapeseed oils depend on higher concentrations of α-linolenic acid C18:3n3 and oleic acid C18:1n9c as variables. In our research, we used PCA analysis to discriminate VO types by FAs. Other authors [71–73] used PCA to reveal factors of the FA composition of various types of VOs.

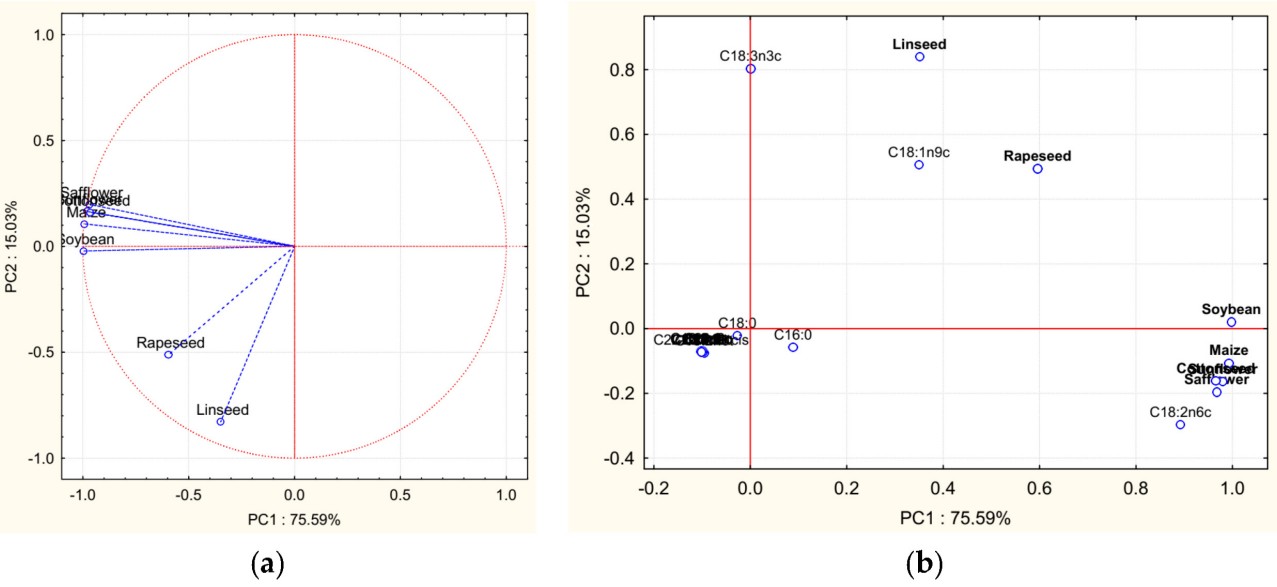

(**a**)　　　　　　　　　　　　　　(**b**)

**Figure 5.** Principal component analysis for FA compositions: (**a**) seven vegetable oils were displayed in loading plot; (**b**) PCA loading plot of FA compositions bi-plot.

### 3.8. Artificial Neural Network

In our research, ANN was carried out to classify the samples of VOs by FA composition. As the input layer could consist of FAs as independent variables, in our work, 18 FA compound datasets were used as the input layer, as shown in Figure 6. The seven types of VOs' FA compounds were subsequently classified into three categories, which corresponded with three hidden layers (in Figure 5, marked as H1, H2, H3). Finally, the output layer consisted of VO type. The structure of ANN is 18:3:1. Our ANN analysis used error functions like the sum of squared errors (SSEs) and root-mean-square error (RMSE), as shown in Supplementary Materials (Table S1). In the process of predicting the FA compounds, the regression coefficients were calculated as 0.99, except for cottonseed oil (0.96). According to the Supplementary Materials (Table S1), the RMSE coefficients of sunflower, maize, linseed, soybean and rapeseed oils varied between 0.004 and 0.08, whereas the RMSE coefficients of safflower and cottonseed oils were higher, at 0.10 and 0.20, respectively. According to the Supplementary Materials (Figures S3 and S4), all the VOs along with other types of oils had negative profiles. ANN analysis is widely used to classify and predict VOs by FA composition [74,75]. The obtained FA data can be used for optimizing the pretreatment parameters of the analysis to achieve a higher value-added biodiesel [76].

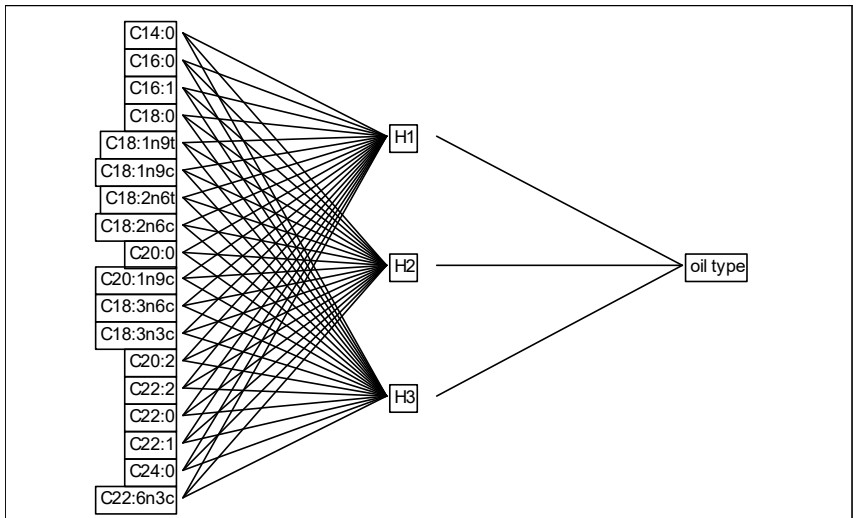

**Figure 6.** Network architecture of the FA compound.

### 4. Conclusions

Kazakhstan's fat and oil products today have great prospects for growth in the foreign market. This research investigated the seven main types of edible VOs produced in Kazakhstan. The FA contents of all VOs produced in Kazakhstan showed differences in the contents of SFAs, MUFAs and PUFAs. Cottonseed oil (25.57%) showed the highest content of SFAs, while rapeseed oil (66.31%) showed the highest content of MUFAs due to the high content of oleic acid. Safflower oil (linoleic acid 74.25%) and linseed oil (α-linolenic 56.01%) showed the highest PUFA content. High contents of α-linolenic C18:3n3 as the main FA in linseed oil presented a low omega-6/omega-3 ratio (0.28%), which is the ideal ratio recommended by the World Health Organization. Because of the richer content of linoleic acid C18:2n6c and the poorer contents of α-linolenic acid C18:3n3 and docosahexaenoic acid C22:6n3, maize oil, cottonseed oil, sunflower oil and safflower oil showed higher omega-6/omega-3 ratios of 103.93, 207.14, 442.86 and 530.35, respectively.

**Supplementary Materials:** The following supporting information can be downloaded at: https://www.mdpi.com/article/10.3390/app13137910/s1, Figure S1. GC chromatogram of the separate FAs of some of the vegetable oils analyzed: (a) safflower; (b) sunflower; (c) linseed; (d) cottonseed. These were analyzed on a high-polarity capillary column CP-Sil 2560. The column's temperature was programmed as follows: 100 °C (5 min), 40 °C/min to 210 °C (8 min) and 10 °C/min to 240 °C (16.5 min). All chromatograms are presented with the same scale for both the x- and y-axes. Figure S2. Correlations between the main fatty acid ratios. Correlations were performed by identifying the Pearson's correlation coefficient. Red indicates a positive correlation, and blue indicates a negative correlation. Table S1. ANN error functions calculated prediction parameters. Figure S3. The predictive capability plot for FA composition of vegetable oils: (a) safflower; (b) sunflower; (c) maize; (d) linseed; (e) cottonseed; (f) soybean; (g) rapeseed oils. Figure S4. Predictive residual plot for FA composition of vegetable oils: (a) safflower; (b) sunflower; (c) maize; (d) linseed; (e) cottonseed; (f) soybean; (g) rapeseed oils. Figure S5. Average values of the SFA, MUFA and PUFA groups (percentage of total FAs) in the vegetable oils from different sources: (a) safflower; (b) sunflower; (c) maize; (d) linseed; (e) cottonseed; (f) soybean; (g) rapeseed oils.

**Author Contributions:** Conceptualization, M.T., M.N. and A.S. (Assiya Serikbayeva); methodology, M.T. and A.S. (Assiya Serikbayeva); formal analysis, M.T., A.S. (Aksholpan Shokan) and N.M.; investigation, Z.S. and K.M.; data curation, M.T., M.N. and A.S. (Assiya Serikbayeva); writing—original draft preparation, M.T. and M.N.; writing—review and editing, A.S. (Assiya Serikbayeva) and M.N.; visualization, M.T. and A.S. (Aksholpan Shokan); supervision, M.N.; project administration, A.S. (Assiya Serikbayeva); funding acquisition, A.S. (Assiya Serikbayeva). All authors have read and agreed to the published version of the manuscript.

**Funding:** This research was funded by the Ministry of Agriculture of the Republic of Kazakhstan via the project "Development of technologies for processing agricultural raw materials using Halal standards" and as part of the scientific and technical program BR10764970 "Development of science-intensive technologies for the deep processing of agricultural raw materials in order to expand the range and yield of finished products from a unit of raw materials" of 2021–2023.

**Institutional Review Board Statement:** Not applicable.

**Informed Consent Statement:** Not applicable.

**Data Availability Statement:** Not applicable.

**Conflicts of Interest:** The authors declare no conflict of interest.

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
