# Peer review of "Comparative Analysis and Determination of the Fatty Acid Composition of Kazakhstan’s Commercial Vegetable Oils by GC-FID"

_applsci, doi:10.3390/app13137910_

Round 1
Reviewer 1 Report
Page-2 Line 45 what is uniqueness of your raw fat
Page-2 line 66 what is full name FA
page -2 line 80 rewrite the sentences
the authors focus on last paragraph of introduction. Describe more detail of your research out put in last part of introduction
why the share of sunflower decreased in the period of 2015-20200although its production increased during this periods
Tables should ne unified in whole manuscript
focus overall English quality
There are many spilling and grammatical errors. these should be improved in revision.
Author Response
Point 1: Page-2 Line 45 what is uniqueness of your raw fat
Response 1: Comment accepted and text corrected. Most of oilseed crops cultivated for local environmental conditions, which could influence to fatty acid contents. In revised manuscript added Table 4, where shown comparison fatty acid contents from other researchers and differences each oil types.
Point 2: Page-2 line 66 what is full name FA.
Response 2: This was fatty acid (FA). Specific verb were added in Abstract.
Point 3: The authors focus on last paragraph of introduction. Describe more detail of your research out put in last part of introduction.
Response 3: There added more detail about our research. Especially, the purpose of this research was analyze fatty acid compositions, which gave more information about produced edible oil in local conditions.
Point 4: Why the share of sunflower decreased in the period of 2015-2020, although its production increased during this periods.
Response 4: Comment accepted and text corrected. In 2015-2020, the production of sunflower oil increased, but the percentage decreased from 77% to 70%, due to the rapid growth of other oils.
Point 5: Tables should ne unified in whole manuscript.
Response 5: Comment accepted and all tables unified.
Point 6: Focus overall English quality.
Response 6: English language has been improved before submission, using MDPI language editing service (certificate english-65979).

Reviewer 2 Report
This manuscript presents the analysis of seven edible vegetable oils produced in Kazakhstan. After reading the document carefully, I think the authors may address the following issues to strengthen the manuscript:
1. What is the purpose of analyzing these oils for adulteration? To establish the composition for producing higher value-added products such as biodiesel? The authors should clarify this point in the introduction of the manuscript.
2. The authors should discuss the similarity between the oils analyzed and other studies previously conducted by other authors. Including a table comparing the samples studied with others previously published could be interesting.
3. The point "2.1. Data Collection" I think should be included in the article's introduction and not in the methodology.
4. Include how SFA, MUFA and PUFA were calculated in the experimental part.
5. Samples", the authors indicate that they purchased from supermarkets a total of 35 samples: sunflower oil (n=9), linseed oil (n=7), maize oil (n=4), safflower oil (n=3), cottonseed oil (n=3), soybean oil (n=6) and rapeseed oil (n=3). But in point "3.2. Determination of FA in vegetable oils", they indicate that seven commercially available vegetable oils were analyzed. The authors should specify whether they analyzed the 35 original samples. If so, they should give the analysis of each of the samples studied. If only seven samples were analyzed, they should indicate how they were selected from the 35 samples initially acquired.
Author Response
Point 1: What is the purpose of analyzing these oils for adulteration? To establish the composition for producing higher value-added products such as biodiesel? The authors should clarify this point in the introduction of the manuscript.
Response 1: The main purpose of this research was analyze the fatty acid composition available for other scientists, food and energy industries. These data gave information about FA in edible vegetable oil, which produced and cultivated in local environmental conditions.
Point 2: The authors should discuss the similarity between the oils analyzed and other studies previously conducted by other authors. Including a table comparing the samples studied with others previously published could be interesting.
Response 2: In revised manusript was added Table 4, where shown comparative study our fatty acid researches with other scientists sources. There were chosen researches, where studied commercially edible oils.
Point 3: The point "2.1. Data Collection" I think should be included in the article's introduction and not in the methodology.
Response 3: Comments accepted. Production edible oils in Kazakhstan was described in line 54-57. The table and chart were moved to the Results section, as they were collected as a result of processing the National Bureau statistics website of the Republic of Kazakhstan.
Point 4: Include how SFA, MUFA and PUFA were calculated in the experimental part.
Response 4: SFA, MUFA and PUFA calculation formula added in 3.3. Determination of FA in vegetable oils.
Point 5: Samples", the authors indicate that they purchased from supermarkets a total of 35 samples: sunflower oil (n=9), linseed oil (n=7), maize oil (n=4), safflower oil (n=3), cottonseed oil (n=3), soybean oil (n=6) and rapeseed oil (n=3). But in point "3.2. Determination of FA in vegetable oils", they indicate that seven commercially available vegetable oils were analyzed. The authors should specify whether they analyzed the 35 original samples. If so, they should give the analysis of each of the samples studied. If only seven samples were analyzed, they should indicate how they were selected from the 35 samples initially acquired.
Response 5: There were analyzed 7 type of vegetable oils. Also, there were added number of analyzed each vegetable oils type in Table 3, where calculated mean and standard deviation for each vegetable oil type.

Reviewer 3 Report
This manuscript describes the use of neural networks and GC-FID to detect and distinguish various oils. The research is very interesting. The introduction does a good job of covering the topic. The materials and methods section provides great details on how to reproduce the results. The results are presented clearly, and discussed in light of relevant research. The conclusions are supported by the results. Overal this is a solid contribution to the field.
The authors go to great detail to describe the machine learning technology employed in the manuscript. This is of great benefit to the audience of the journal. The figures provide great detail on the schemes and plans of the research. Overall this is an excellent contribution. A large amount of data is provided, that will aid researchers in evaluating oils from the region.
Specific comments to be addressed prior to publication.
1) The authors should revise the paragraph structure. Many of the paragraphs consist of only a few sentences which is confusing. The authors could expand the paragraphs for more complete discussion. Alternatively, paragraphs could be combined.
2) The authors could go into additional details as to why the neural network methods were used in this study.
Author Response
Point 1: The authors should revise the paragraph structure. Many of the paragraphs consist of only a few sentences which is confusing. The authors could expand the paragraphs for more complete discussion. Alternatively, paragraphs could be combined.
Response 1: Comments accepted and paragraphs combined.
Point 2: The authors could go into additional details as to why the neural network methods were used in this study.
Response 2: Artificial neural networks section has been improved. ANN method used for possible adulteration. Also, obtained fatty acid composition data will help to establish for producing higher value-added products.
